# Inertial Sensor-to-Segment Calibration for Accurate 3D Joint Angle Calculation for Use in OpenSim

**DOI:** 10.3390/s22093259

**Published:** 2022-04-24

**Authors:** Giacomo Di Raimondo, Benedicte Vanwanseele, Arthur van der Have, Jill Emmerzaal, Miel Willems, Bryce Adrian Killen, Ilse Jonkers

**Affiliations:** Department of Movement Sciences, Katholieke Universiteit Leuven, 3001 Leuven, Belgium; benedicte.vanwanseele@kuleuven.be (B.V.); tuur.vanderhave@kuleuven.be (A.v.d.H.); jill.emmerzaal@kuleuven.be (J.E.); miel.willems@kuleuven.be (M.W.); bryce.killen@kuleuven.be (B.A.K.); ilse.jonkers@kuleuven.be (I.J.)

**Keywords:** 3D joint kinematics, joint angles, lower-body kinematics, sensor-to-segment calibration, IMU, wearable sensors, musculoskeletal, biomechanical model, open-source, motion analysis

## Abstract

Inertial capture (InCap) systems combined with musculoskeletal (MSK) models are an attractive option for monitoring 3D joint kinematics in an ecological context. However, the primary limiting factor is the sensor-to-segment calibration, which is crucial to estimate the body segment orientations. Walking, running, and stair ascent and descent trials were measured in eleven healthy subjects with the Xsens InCap system and the Vicon 3D motion capture (MoCap) system at a self-selected speed. A novel integrated method that combines previous sensor-to-segment calibration approaches was developed for use in a MSK model with three degree of freedom (DOF) hip and knee joints. The following were compared: RMSE, range of motion (ROM), peaks, and R^2^ between InCap kinematics estimated with different calibration methods and gold standard MoCap kinematics. The integrated method reduced the RSME for both the hip and the knee joints below 5°, and no statistically significant differences were found between MoCap and InCap kinematics. This was consistent across all the different analyzed movements. The developed method was integrated on an MSK model workflow, and it increased the sensor-to-segment calibration accuracy for an accurate estimate of 3D joint kinematics compared to MoCap, guaranteeing a clinical easy-to-use approach.

## 1. Introduction

The assessment of locomotor function needs to be taken out of the lab and into the patient’s ecological environment to evaluate the true impact of specific clinical treatments such as gait retraining or pain medication on joint function. To do so, joint kinematics and ultimately loading should be accurately measured in real-life conditions. An inertial capture (InCap) system combined with a musculoskeletal (MSK) model that represents the complex kinematics patterns and the mechanical interactions of the joint can be used for instance to evaluate patients’ knee adaptations to instructed gait pattern modifications (toe-in/out, trunk leaning), aiming to decrease knee loading and eventually contributing to intermediate pain alleviation [1,2,3,4,5] and slowing down degenerative joint disease progression in the long term.

Laboratory-based 3D motion capture (MoCap) combined with in-ground embedded force plate systems has long been considered the gold standard method for capturing human movement and musculoskeletal function. Consequently, the use of 3D motion capture data as an input for MSK modelling workflows has become an attractive method to allow the estimation of different movement parameters (e.g., muscle forces and joint contact forces), however restricting the evaluation of locomotor function to highly controlled laboratory settings. In the context of degenerative joint diseases, the validity of MoCap based methods for the evaluation and the analysis of human movements may not reproduce real-life movement patterns. Indeed, laboratory-based methods that document locomotor function represent a limited view of an individual patient’s condition and the need for more ecological studies of locomotor function are being advocated [6]. 

Recently, wearable sensor technologies have gained massive development, with increased performance due to reduced weight and size, long battery life, and versatility [7]. In particular, InCap systems have become an attractive alternative to 3D MoCap systems, as inertial measurement units (IMUs) are capable of estimating the orientation of 3D sensors and segments and by extension 3D joint kinematics [8,9,10]. Previous studies have compared the accuracy of the InCap and the MoCap systems based on sagittal, frontal, and transverse plane movement kinematic estimation, reporting errors between 5° and 18° for sagittal and non-sagittal plane angles, respectively [11,12]. In most clinical applications, an error less than 2° is considered acceptable, whereas errors between 2° and 5° are also acceptable but require specific interpretation [11,13].

Based on the current limiting accuracy, especially for non-sagittal plane kinematics, it is crucial to improve the accuracy of InCap 3D joint kinematic estimations to reach a similar level of accuracy as seen in 3D MoCap. Such improvements would provide clear applications in several MSK disorders (e.g., osteoarthritis (OA) and cerebral palsy), where the joints range of motion (ROM) or movement extremes are necessary to properly analyze and monitor joint loading in clinical practice [14,15].

Kinematics parameters such as gait speed, stride length, or joint angles in patients with osteoarthritis (OA) can discriminate the OA population from healthy subjects [16]. Further, knee osteoarthritis (KOA) patients in particular exhibit significant reductions in knee flexion ROM (6–10°) and peak knee flexion (10–15°) [17,18,19], as well as a significant increase in knee adduction ROM (2–5°) and maximal peak (5–10°) during walking compared to healthy subjects [20,21]. Moreover, these results showed that the altered range of motion of the affected joint (hip, knee) led to a compensatory increase in pelvic motion (list 5–8°), thereby affecting the lumbar joint and contributing eventually to back pain [18]. Furthermore, typical modifications in joint kinematics introduced during gait retraining, 5–10° toe-in, toe-off, or trunk lean in KOA patients led to positive changes in order of 1–3° for pelvic tilt and list, 2–5° for knee adduction, and 5–10° for hip adduction and rotation. These gait pattern modifications can reduce knee moments and knee loading during gait with the potential to slow disease progression [4,5,22,23]. The ability of current InCap systems to detect such clinically relevant difference is limited, making improvements to InCap estimation of kinematics even more important for clinical applications. 

The main factor limiting kinematic accuracy using InCap systems is the sensor calibration [24]. InCap sensor-to-segment calibration is crucial to estimate the body segment orientations from the sensor orientations and consequently perform an accurate estimation of 3D joint kinematics [25,26]. In past years, several sensor-to segment calibration methods have emerged. The *manual calibration* procedure utilizes manual alignment of the sensor reference system axes with the relative body segment axes [24]. This method is preferred in rehabilitation settings due to usability, speed, and being user-friendly; however, the reliability and the repeatability of this method is entirely examiner dependent. A second approach is *static calibration* procedures, which use a sensor’s gravity vector aligned with the appropriate body segment axes during static poses such as the N-pose or the T-pose [27]. This method presented root mean square errors (RMSE) < 6° and correlation between InCap and MoCap (r) > 0.8 for sagittal angles (hip rotation RMSE > 6°) [8,28,29,30]. Nonetheless, static calibration methods rely on the assumption that the subject can naturally adopt a specific posture, with the body segments in an anatomically neutral position, aligned or perpendicular to the vertical gravity vector which may be difficult for specific pathological conditions (e.g., knocked knee alignment). The static calibration procedure was recently implemented in an open-source toolbox (OpenSense) that provided a guided procedure to calibrate IMU sensors with a MSK model [31] informed by the subject standing pose recorded through 3D MoCap. Joint angle estimates based on this toolkit provided acceptable results, especially for sagittal plane kinematics with a median RMSE between 3–6° for the hip and knee sagittal plane movement and correlations from 0.36–0.76 [32]. However, frontal and transverse plane angle kinematics such as knee adduction, rotation, and ankle pronation were not evaluated. Furthermore, MoCap data is still required for defining the initial pose therefore limiting its use as a pure substitute for MoCap. A *third* calibration method is a *functional calibration* method that assumes a perfect alignment of the sensor and the joint rotation axes during specific movements (e.g., hip flexion-extension, knee flexion-extension, sit-to-stand, walking, and circling movement). This method has proven to be more appropriate for estimating frontal and transverse angles. For example, methods based on hip adduction or cycling motion show errors below 3.8° for sagittal plane angles and below 6° for non-sagittal plane angles during walking and cycling [33,34,35,36,37]. However, the assumption of pure rotational movement around the segment axes during calibration movement, imposes specific challenges in patients with a limited range of motion. *The final method is anatomical calibration* which uses external devices with integrated sensors to locate the joint axes with respect to the sensor axes system [38,39]. This anatomical calibration method shows errors <4° for sagittal and non-sagittal angles except for ankle pronation (9.7°) during walking [40]. Although anatomical calibrations require supplementary tools and trained operators to properly locate the landmarks, this method can be applied to patients with limited joint mobility. Overall, only a few studies [35,41] combined static and functional calibration methods and reported RMSE < 5° for sagittal plane and non-sagittal plane angle during walking only.

Advanced algorithms have been developed to overcome other current calibration accuracy limitations such as soft tissue motion artifacts caused by no-rigid connection between the body and the sensors [42,43] and magnetic distortions [9] caused by ferromagnetic objects that may affect the IMU sensor’s orientation. For instance, closed-source commercially available models, such as the Xsens system can successfully compensate for magnetic distortions and drifts based on an in-house anatomical model and algorithms during any motion with an easy-to-use calibration strategy [27,44]. However, an open-source platform to compute 3D joint kinematics from IMU signals (independent from the InCap system used) for any type of motion, with no need of MoCap, would be highly beneficial, and it is currently unavailable.

This study aims to (1) develop an integrated functional sensor-to-segment calibration open-source workflow applicable in a clinical context and therefore independent of 3D MoCap for computing three-dimensional pelvis, torso, and lower limb joint kinematics (hip, knee, ankle) with IMU sensors based on an MSK model in the open-source software OpenSim; and (2) evaluate its accuracy compared to the gold standard MoCap system during walking, stair ascent and descent, and running. Ideally, the accuracy of such a workflow would allow estimation of kinematics within an accuracy of ±5°, which allows discriminating joint kinematic of healthy and OA subjects, and it would allow for the measuring of differences in joint kinematics targeted during gait retraining. Furthermore, performance would be expected to be similar during a variety of different relevant, functional movements.

## 2. Materials and Methods

### 2.1. Data Collection

3D MoCap data was collected at the Movement & Posture Analysis Laboratory Leuven (MALL), at the Department of Movement Science, KU Leuven, Leuven, Belgium. This research was in accordance with the ethical guidelines provided by the ethical research committee KU Leuven (reference no. G-2021-3436). Eleven healthy adults with no musculoskeletal or neurological disorders volunteered for the study (6 males and 5 females; age: 26.0 ± 3.1 yr; height: 1.77 ± 0.9 m; weight 68.4 ± 9.9 kg). All participants provided written informed consent, prior to data collection.

### 2.2. Instrumentation

Full-body InCap data were collected using the Xsens MVN Awinda system (Xsens Technologies B.V., Enschede, The Netherlands), for which 17 IMUs were positioned based on the manufacturer guidelines (MVN User Manual). To record the motion at 60 Hz, Xsens MVN Analyze Pro 2021.0 was used. Simultaneously, a 10 infrared camera MoCap system (VICON, Oxford Metrics Group, UK) was used as the gold standard reference system to record the 3D position of the 54 reflective markers attached on anatomical landmarks of the different body segments (Plug-in Gait Reference Guide—Nexus 2.12 Documentation—Vicon Documentation) at 120 Hz (Appendix A and Appendix A). Both systems were time synchronized based on the manufacturer’s guidelines with a specific trigger at the start/stop recording time.

### 2.3. Protocol

3D InCap and MoCap data were collected, while each subject performed four barefoot movement tasks at self-selected speed: (a) 10 m of over ground walking, (b) 10 s of treadmill running, (c) ascending, and (d) descending stairs. For calibration, participants started each task with specific postures/movements: (i) N-pose, standing with arms to the sides, feet shoulder width apart, and facing forward for a period of five seconds, (ii) sitting in a wooden chair facing forward, (iii) knee extension followed by knee flexion for each leg (iv) standing in N-pose for three seconds, (v) hip abduction/adduction movement (three repetitions for each leg) with no movements in knee or ankle joints; the latter movement execution enforced similar angular velocities of the three segments—thigh, shank and foot, and (vi) participants performed the tasks (a—b—c—d). Raw data recorded by the InCap and the MoCap system were exported to .mvnx and .c3d files using Xsens MVN Analyzer Pro 2021.0 and Nexus 2.12, respectively, and they were imported into MATLAB (R2020a, MathWorks). For both systems, all movements were time-normalized (101 data points at time intervals of 1% of a gait cycle) and processed using custom-built MATLAB scripts.

### 2.4. Kinematics Analysis

3D MoCap kinematics were calculated using the inverse kinematics tool in OpenSim v4.2 using an MSK model with 21 segments and 41 degrees-of-freedom (Hamner 2010 [45]). The knee joint was modeled as a ball-and-socket joint (3 degrees-of-freedom). 3D InCap kinematics were calculated using the inverse kinematics tool in the OpenSense toolkit [32], with a newly developed and integrated sensor-to-segment calibration. 

### 2.5. Sensor-to-Segment Calibration

This integrated sensor-to-segment calibration method (method 3—M3) can be used independent of 3D MoCap data, and it relies on a combination of previously published methods [33,46,47,48] (Figure 1 and Figure 2). Firstly, functional calibration of thigh, shank, and foot sensors was based on analytical hip abduction–adduction motion [33] to improve the estimation of sagittal and non-sagittal plane knee and ankle kinematics (method 1—M1). The reference frame misalignment of the thigh and the shank sensors was estimated by comparing the angular velocity vectors during a hip ab-adduction movement. Each sensor’s angular rotation along the global vertical axis was estimated given that the angular velocity of the thigh and the shank sensors should be identical, as during the hip ab-adduction, the knee motion can be assumed to be approximately zero. Therefore, the thigh and the shank can be considered as a single segment with identical angular velocity. This procedure was then extended and applied to the foot segment. Secondly, functional calibration of the torso, pelvis, and thigh sensors was based on a walking and a sit-to-stand motion (STS) [48] to improve the estimation of sagittal and non-sagittal plane lumbar, pelvic, and hip kinematics (method 2—M2). Torso and pelvis sensors were aligned based on STS motion, whereas thigh sensors were aligned based on the walking motion under the assumption that these movements occur mainly around the global medio-lateral axis. To this end, principal component analysis (PCA) was used to determine the sensor rotation that maximized the movement around the medio-lateral axis. Finally, the initial pose of the model segments was estimated using the approach presented in Fasel et al. [47], with adaptations regarding the utilized OpenSim MSK model and Xsens coordinate reference systems [31,49].The segment global reference frame was defined with the vertical axis aligned with gravity vector, the anterior–posterior axis perpendicular to the gravity vector and pointing in the MSK model forward direction and the medio-lateral axis defined as the cross-product between the vertical and the anterior–posterior axes, pointing to the right of the MSK model. It was assumed that the torso, pelvis, and lower limb segments were orientated in the same direction while standing. The segments’ inclinations were determined using the accelerometer data and the gravity vector. The initial joint angles of the MSK model were then computed based on the segment orientation.

Performance of the developed method M3 (combination of M1 and M2) was evaluated against the anatomical calibration that uses an anatomical position as an initial guess for the standard static calibration with the OpenSense toolkit (M0). Specifically, the anterior–posterior axis of the pelvis sensor was aligned with the anterior–posterior axis of the pelvis segment of the MSK model; then, torso, thigh, shank, and foot sensors were assumed to be aligned with that of the pelvis. For the calibration reference system, segment and joint coordinate systems were defined according to ISB recommendations [50,51].

### 2.6. Statistics

For multiple comparisons, the Wilcoxon rank test was implemented in MATLAB, and it was used to statistically assess significant differences in the root mean squared error (RMSE), the determination coefficient R^2^, range of motion (ROM) difference, and the maximum peak difference between each calibration method. Significance was set at *p* ≤ 0.05. A false discovery rate (FDR) statistical approach in multiple assumptions testing was used for multiple comparisons in order to correct for random events that falsely appear significant. A q-value threshold of 0.05 (FDR of 5%) among all significant variables was used. To evaluate the functional relevance of the observed differences between methods for the different joints, the error in kinematics between different InCap methods and MoCap was evaluated against reported differences in kinematics between control and OA subjects [52,53,54,55].

## 3. Results

Complete tables and figures for comparison of all sagittal and non-sagittal plane kinematics are presented in the Appendix A.

### 3.1. Walking

Knee: Figure 3 shows a representative result of estimated knee angles with the integrated method M3 during the different movements. During walking, M3 significantly reduced errors in estimated frontal and transverse average plane offset compared to the other methods: M0 (8°), M1 (6°), M2 (4°), M3 (2°). These offsets with respect to MoCap angular values can be noticed at the start of the curves in Figure 3. For M3, the overall average RMSE of the knee flexion, abduction, and internal rotation angle were 3.2 ± 1.2°, 2.5 ± 2.2°, and 4.5 ± 1.7°, respectively, with average R^2^ > 0.7 ± 0.2. ROM errors were 1.1 ± 0.6°, 2.2 ± 1.8°, and 2.9 ± 2.2°, and peak magnitude errors were 1.6 ± 2.5°, 3.2 ± 2.1° and 4.3 ± 2.7° for knee flexion, abduction, and internal rotation angle, respectively (see Table 1 and Table 2). M3 showed the highest accuracy compared to the other InCap methods. Where average RMSE reduced significantly from 7.4 ± 2.4° to 2.1 ± 2.4° and from 8.4 ± 3.5° to 4.3 ± 2.0°, for knee adduction and rotation, respectively, compared to M0. Compared to MoCap, significant differences were shown in peak knee adduction and rotation in M0 and between M1 and M3 for knee rotation. RMSE averaged over all joint planes improved for M3: 3.2 ± 1.6° compared to M1: 5.0 ± 2.4° and M2: 5.1 ± 2.8°.

The RMSE between kinematics estimated from MoCap and the different methods (M0–M1–M2–M3) during walking are shown in Figure 4. M3 showed an average RMSE < 5° for all joints and DOFs excluding ankle pro/supination and pelvis internal/external rotation. M0 showed an average RMSE above 5° for all joints, M1 < 5° for hip and knee adduction and M2 < 5° for hip and knee flexion. Statistically significant differences were found between MoCap and InCap in M0 for hip adduction, knee adduction, knee rotation, pelvic tilt, and pelvic rotation during walking, with similar trends during running, and the ascent and the descent of stairs.

Similar to the knee, the integrated method M3 significantly decreased on average the estimated angle offset error for all the other joints observed—lumbar, pelvic, hip, and ankle joint—compared to the other methods: M0 (10°), M1 (8°), M2 (4°), M3 (2°)—(Appendix A) during walking.

Hip: M3 showed the highest accuracy compared to the other InCap methods, with average RMSE being significantly reduced from 7.1 ± 3.1° to 3.2 ± 1.9°, for hip non-sagittal plane angles (Appendix A) compared to M0, with average R^2^ > 0.7 ± 0.2. ROM errors were 3.1 ± 2.6°, 3.6 ± 1.8°, and 3.3 ± 3.1° for hip flexion, hip adduction, and hip rotation, respectively. The errors for the peak were 3.2 ± 1.1°, 3.5 ± 2.3°, 4.0 ± 3.3°. Compared to MoCap, significant difference was found in the peak hip rotation in M0 and between M0 and M3. The RMSE averaged over all joint-planes improved for M3: 3.4 ± 1.2° compared to M1: 5.6 ± 3.5° and M2: 5.2 ± 2.8°.

Lumbar: M3 showed the highest accuracy compared to the other InCap methods, with average RMSE being significantly reduced from 9.1 ± 4.0° to 3.9 ± 2.2° over all joint-planes compared to M0, with average R^2^ > 0.8 ± 0.2. ROM errors were 1.4 ± 2.2°, 2.9 ± 3.9° and 2.9 ± 2.1° for lumbar extension, bending, and rotation, respectively. The errors for the peak were 3.4 ± 3.2°, 3.1 ± 2.7° and 2.2 ± 2.4°. Compared to MoCap no significant difference was found between methods. RMSE averaged over all joint-planes improved for M3: 3.9 ± 2.2° compared to M1: 8.8° ± 3.5° and M2: 6.7 ± 3.2°.

Pelvic: M3 showed the highest accuracy compared to the other InCap methods with average RMSE being significantly reduced from RMSE 10.5 ± 3.7° to 4.0 ± 1.3°, over all joint planes compared to M0, with average R^2^ > 0.7 ± 0.3. ROM errors were 1.3 ± 1.5°, 3.2 ± 1.7° and 2.8 ± 1.7° for pelvic tilt, list, and rotation, respectively. The errors for the peak were 2.6 ± 2.0°, 1.6 ± 1.6° and 2.2 ± 2.4°. Compared to MoCap significant difference was found for pelvic rotation in M0, M1, and between M2 and M3. RMSE averaged over all joint planes improved for M3: 4.0 ± 1.3° compared to M1: 8.0 ± 4.0° and M2: 4.2 ± 1.8°.

Ankle: M3 showed the highest accuracy compared to the other InCap methods with average RMSE being significantly reduced from RMSE 7.3 ± 3.8° to 4.8 ± 1.9° over all joint planes compared to M0, with R^2^ > 0.8 ± 0.2°. ROM errors were 3.7 ± 2.9° and 3.1 ± 1.5° for dorsi-plantar flexion and prono-supination. The errors for the peak were 4.1 ± 2.2° and 4.6 ± 2.7°. Compared to MoCap, no significant difference was found. RMSE averaged over all joint planes improved for M3: 4.8 ± 1.9° compared to M1: 5.6 ± 3.5° and M2: 5.0 ± 3.1°.

### 3.2. Running and Stair Ascent and Descent

For running (Appendix A), accuracy relative to Mocap was lower compared to walking across all methods. In particular, in non-sagittal plane angles, M3 showed for hip, knee, and ankle frontal and transverse plane angles on average a RMSE 5.1 ± 2.9° and 5.2 ± 2.1° compared to 3.5 ± 1.9° and 4.0 ± 1.5° during walking. Similarly, stair ascent and descent movement (Appendix A) showed smaller accuracy compared to walking across all methods. In particular, lumbar and pelvic angles during stair ascent showed in sagittal, frontal and transverse planes on average a RMSE 5.5 ± 3.2°, 5.3 ± 2.2° and 5.2 ± 2.1° compared to 3.5 ±1.8°, 3.0 ± 0.9° and 4.1 ± 1.8° during walking (Appendix A).

## 4. Discussion

### Summary and Main Findings

The aim of this study was to develop an integrated functional sensor-to-segment calibration applicable in a clinical context for computing three-dimensional pelvis, torso, and lower limb joint kinematics (hip, knee, ankle) during common daily life activities based on IMU sensors combined with an OpenSim MSK model. The developed method addresses previously documented challenges within literature, specifically sensor-to-segment calibration to estimate the body segment orientations from the sensor orientations without MoCap data and the integration with a complex MSK modeling workflow to estimate 3D joint kinematics [15,32,56,57]. In general, the integrated method based on the combination of PCA and functional calibration (M3) was for the sagittal, frontal, and transverse joint angles more in agreement with the gold standard MoCap (RMSE< 5°—except transverse plane in ankle joint) than previously reported methods. These results confirm that calibration methods based on functional movements such as hip abduction/adduction, sit-to-stand, and walking improved the sensor-to-segment accuracy, particularly for non-sagittal plane kinematics.

In contrast to previous studies that only focused on specific activities [28,35,58], the integrated method (M3) can be used for evaluating different daily life movement tasks, such as over ground walking, running, and stair ascent/descent. This method does not require specific subject postures, the presence of expert operators, nor the use of external devices as some previous works proposed [38,40,59,60]. Moreover, the integrated method (M3) shows overall more accurate kinematics compared to previous calibration methods (i.e., M0, M1, M2 based on previous works) [33,36,46]. In fact, this method measures estimates of sagittal and non-sagittal plane kinematics that are only less than 5° difference compared to MoCap during walking (where errors between 2°–5° are considered clinically acceptable). The M0 method based on anatomical/static calibration (M0) showed the lowest accuracy, particularly for non-sagittal plane angles, indicating that functional methods are required to improve the non-sagittal plane kinematic estimations. The method based only on hip ab-adduction calibration (M1) showed good accuracy for non-sagittal plane angles comparable with previous works [34,61] during walking but lower accuracy in sagittal plane angles, especially during stair ascent and descent. The walking based calibration method (M2) showed good accuracy for sagittal plane angles in agreement with previous work [48] but lower accuracy in non-sagittal plane angles across all motions. Therefore, method M3 was the most reliable method for accurate sensor-to-segment calibration, resulting in computing sagittal and non-sagittal angles joint kinematics with an error <5° for out of the lab contexts compared to MoCap.

In addition, we evaluated the performance of the integrated method M3 by calculating the differences in ROM and in angle peaks compared to MoCap. The ΔROM and Δpeak error of sagittal and non-sagittal plane angles varied between 2° and 5°. Such ROM and peak differences have been demonstrated to be important parameters for a variety of clinical applications [62,63]. For instance, ROM in knee osteoarthritis patients was reduced compared to healthy subjects during walking [19,55,64]. M3 accuracy would allow detecting previously reported differences between healthy and OA patients, which are in the order of 6–10° in knee flexion ROM, 5–10° in peak knee adduction, 5–10° in hip adduction and rotation, and 5–8° in pelvic list ROM, which is within the range of accuracy indicated in this preliminary investigation in healthy individuals. However, a detailed analysis and repetition of this research would be required for a dedicate knee OA cohort to confirm this, and it will form the basis of future work [52,53,55]. However, changes in pelvic list, tilt or knee adduction introduced during gait retraining such as toe-in or toe-out in the order of 1–3° for pelvic list and tilt and 2–5° for knee adduction may be difficult to detect, given the M3 average error being 3–5°. Moreover, during stair ascent and descent the error is slightly over 5° compared to MoCap, which in some cases can be considered not clinically acceptable.

Given that the InCap system is an easy-to-use system capable of accurately detecting the difference in ROM of 5°, it is a valid alternative to conventional MoCap systems for the assessment of 3D joint kinematics in clinical and ecological settings. However, its accuracy decreases for frontal and transverse plane angles: in particular, for joints with small ROM such as knee adduction (2–5°), where a RMSE threshold of 5° may represent an error of 100% of ROM. Here, correspondence in terms of correlation of the kinematics may yield a more accurate comparison for evaluating the accuracy of the InCap systems [38,53,65]. Consequently, we are aware that obtaining accurate non-sagittal plane angles with an InCap system during real-life conditions is complex, especially when the differences in ROM and peaks between for instance patient populations or conditions are <5°. Nevertheless, due to the combination of functional calibration methodologies applied for developing the integrated M3, kinematic estimation accuracy improved across all angles and planes, with an ascent to levels comparable to MoCap. Furthermore, our developed method does not rely on manual offset correction as previously used during for instance stair ascent and descent [66,67].

Therefore, for applications where the specific clinical intervention, e.g., gait retraining in patients with knee OA, needs accurate measurement of knee non-sagittal plane angles the developed functional calibration approaches in combination with an MSK model with multiple degrees-of-freedom could be used if future studies confirm similar levels of accuracy in OA patients [68,69]. However, the selected calibration method depends on the degree-of-freedom of the joints in the MSK model. Indeed, an MSK model with one degree-of-freedom for the knee joint and the ankle joint as previously described for use with OpenSense, does not need functional calibration and, in this case, methods based only on anatomical/static calibration are sufficient to accurately estimate sagittal plane joint kinematics [32].

Our study showed that the accuracy of the developed calibration method is higher than previously developed methods—M0, M1, M2—and that the effect is consistent across multiple movement tasks. Nevertheless, there seems to be a negative effect on the calculated error due to the more dynamic movement tasks such as running. This may be due to the limited sample frequency used by the InCap system, which was only 60 Hz. Therefore, future studies should consider a higher sample frequency during such high dynamic movements. Another limitation of the presented method is that it relies on the ability of the subjects to perform specific joint movements—the hip abduction/adduction movement, and the sit-to-stand movement. Therefore, functional calibration may be challenging in subjects with limited joint mobility or impairments. Furthermore, the repeatability of sensor placement was not investigated in this study. Sensor placement repeatability is operator dependent and the introduced misalignment is known to account for up to a 55% error in the sagittal plane angles, while the sensor-to-segment calibration misalignment is known to account for up to 40–50% in non-sagittal plane angles [26,70]. By introducing the proposed sensor-to-segment calibration, sensor misalignment, including that introduced due to between operator variability in sensor placement, is expected to be reduced as this will automatically be accounted for during the functional movements [71]. Moreover, the integrated method was applied during movement tasks with a duration under two-minute motions, hence, it may be necessary to mitigate errors due to drift during movement tasks with a longer duration. However, method 3 may be applied with drift-reduced estimation as presented in the Fasel et al., 2018 study [36]. Drift is estimated and reduced based on the accelerations’ vector in the global frame.

This study impacts the use of wearable sensor technology for documenting 3D knee joint kinematics (e.g., patients with knee OA). However, to evaluate the impact of specific clinical treatments out-of-the-lab such as gait retraining or pain medication, the knee loading should ideally be accurately measured in real-life conditions, e.g., to estimate the impact of the patient’s gait pattern modifications (toe-in/out, trunk lean) on knee loading [2,4,72,73,74,75]. Future work should therefore validate the method developed for the individual activities of daily living with more natural activity sequences, and it should focus on the estimation of kinetics, in particular the ground reaction forces and joint moments estimation to enable true real-world monitoring of the impact of feedback and gait interventions on an individual patient’s locomotor function and joint loading. Such an approach would require valid and reliable methods for the 3D plane angle estimation [76] but also numerical methods to estimate ground reaction forces and moments without force platforms, i.e., solely based on kinematics data combined with a model of the foot-ground contact [77,78]. In this context, machine learning based and probabilistic methods are now being introduced to estimate ground reaction forces and moments using InCap systems [79,80,81,82].

## 5. Conclusions

In conclusion, in this study we developed a novel integrated sensor-to-segment calibration method for IMU sensors that improves non-sagittal plane kinematic estimates in a range of movements. Moreover, keeping the focus on future clinical applications, this method aimed to calculate the 3D kinematics of the torso, pelvis, and lower limbs, allowing for a not-predefined location of the IMU-sensors thereby guaranteeing a clinical easy-to-use approach.

## Figures and Tables

**Figure 1 sensors-22-03259-f001:**
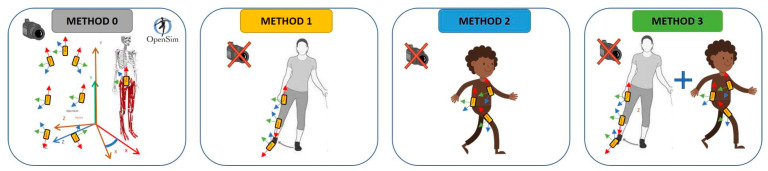
Method 0: One sensor (e.g., pelvis) aligned with the MSK model (MoCap derived pose) and with all the other sensors—static calibration approach [46]; Method 1: hip abduction–adduction motion to align thigh-shank-foot sensors—functional calibration [34]; Method 2: PCA-method based on sit-to-stand and walking to align torso-pelvis-thigh sensors—functional calibration [46]; Method 3: combination of Methods 1 and 2 for the alignment of all sensors—functional calibration [34,46].

**Figure 2 sensors-22-03259-f002:**
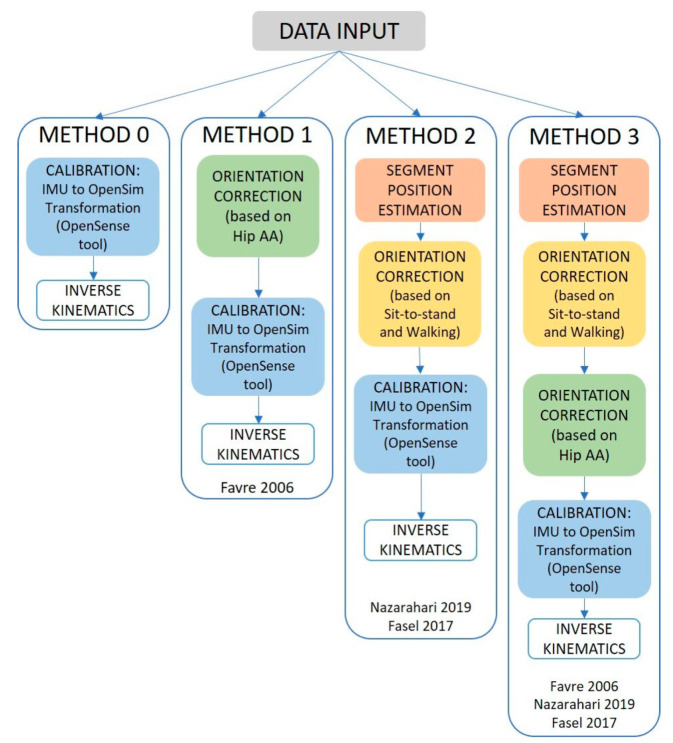
Workflow scheme for the different methods: the green, orange, and yellow boxes are part of the functional calibration and sensors alignment.

**Figure 3 sensors-22-03259-f003:**
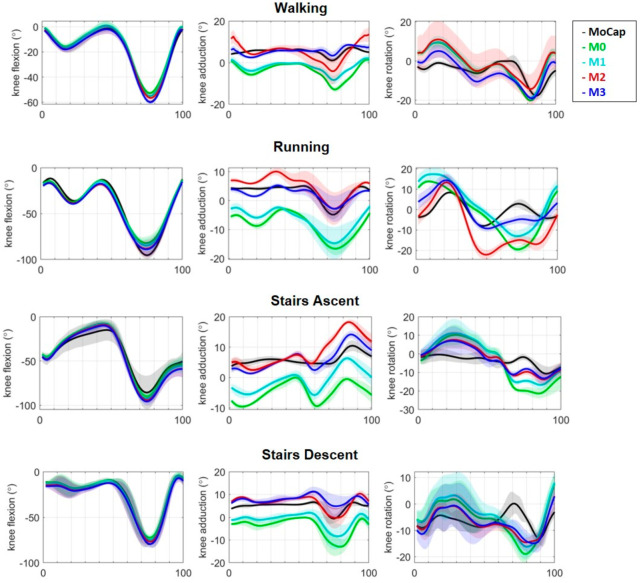
3D knee joint angle comparison during walking, running, stair ascent, and stair descent over a cycle for a representative subject—in order sagittal, frontal, and transverse planes. InCap system (M0—green, M1—cyan, M2—red, M3—blue) and MoCap system (black). M0: OpenSense standard pipeline, M1: functional hip abd-adduction motion calibration, M2: functional PCA walking calibration, M3: functional hip abd-adduction and walking PCA calibration.

**Figure 4 sensors-22-03259-f004:**
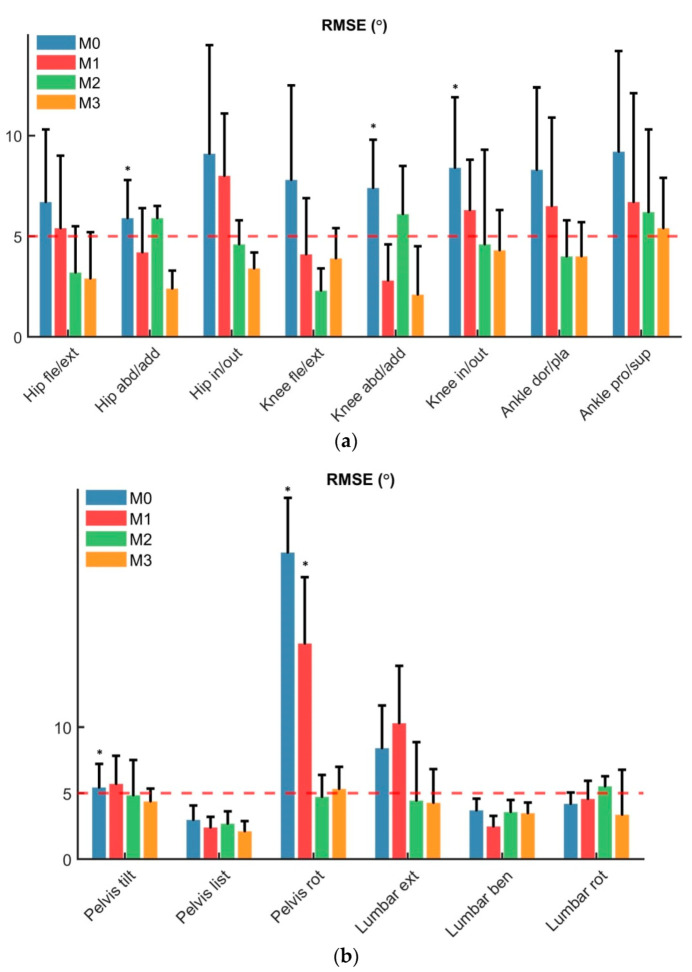
Subjects RMSE (mean ± SD) for InCap joint kinematics during walking compared to clinical threshold of 5.0 deg (red dotted line). Differences between the developed methods (M0—blue, M1—red, M2—green, M3—orange) across the joint-planes (**a**): Hip flexion/extension, hip abduction/adduction, hip internal/external rotation, knee flexion/extension, knee abduction/adduction, knee internal/external rotation, ankle dorsiflexion/plantarflexion, ankle pronation/supination; (**b**): pelvic tilt, pelvic list, pelvic rotation, lumbar extension, lumbar bending, lumbar rotation. * *p* ≤ 0.05. M0: OpenSense standard pipeline, M1: functional hip abd-adduction motion calibration, M2: functional PCA walking calibration, M3: functional hip abd-adduction and walking PCA calibration.

**Table 1 sensors-22-03259-t001:** Values between calibration methods and MoCap during the different motions for knee joint. RMSE: root mean square error, R^2^: coefficient of determinationSD: standard deviation. Significant differences *p* < 0.05—values in bold. W: walking; R: running; S.A. stair ascent; S.D: stair descent.

Knee		RMSE (°)				$R^{2}$			
		Method 0	Method 1	Method 2	Method 3	Method 0	Method 1	Method 2	Method 3
		Mean (SD)				Mean (SD)			
W	Flexion	7.8 (4.7)	4.1 (2.8)	2.3 (1.1)	3.9 (1.5)	0.9 (0.1)	0.9 (0.1)	0.9 (0.1)	0.9 (0.0)
	Adduction	**7.4 (2.4)**	2.8 (1.8)	6.1 (2.4)	2.1 (2.4)	0.1 (0.4)	0.8 (0.5)	0.2 (0.4)	0.7 (0.5)
	Rotation	**8.4 (3.5)**	**6.3 (2.5)**	4.6 (4.7)	4.3 (2.0)	0.3 (0.5)	0.5 (0.5)	0.4 (0.4)	0.6 (0.2)
R	Flexion	**9.8 (4.0)**	4.2 (1.9)	4.6 (2.0)	3.5 (3.0)	0.7 (0.6)	0.9 (0.1)	0.7 (0.6)	0.9 (0.1)
	Adduction	**8.7 (4.5)**	4.7 (3.3)	6.0 (4.9)	5.6 (2.7)	0.2(0.6)	0.8 (0.5)	0.3 (0.5)	0.7 (0.5)
	Rotation	9.5 (4.9)	7.1 (6.0)	6.7 (5.0)	4.8 (3.1)	0.3 (0.6)	0.4 (0.5)	0.3 (0.6)	0.5 (0.4)
S.A.	Flexion	7.8 (5.0)	4.2 (2.9)	4.6 (3.0)	3.5 (2.0)	0.9 (0.1)	0.9 (0.1)	0.9 (0.1)	0.9 (0.1)
	Adduction	**9.7 (4.5)**	4.4 (2.3)	4.0 (2.4)	3.6 (2.1)	0.2 (0.5)	0.3 (0.3)	0.2 (0.5)	0.5 (0.3)
	Rotation	9.0 (3.9)	5.1 (3.0)	6.2 (2.0)	4.8 (3.1)	0.6 (0.4)	0.2 (0.3)	0.4 (0.4)	0.6 (0.3)
S.D.	Flexion	**12 (5.2)**	13 (4.9)	4.6 (1.6)	3.5 (3.1)	0.9 (0.1)	0.9 (0.1)	0.9 (0.1)	0.9 (0.1)
	Adduction	**8.7 (4.0)**	4.2 (2.8)	4.0 (1.9)	4.6 (2.6)	0.3 (0.3)	0.4 (0.3)	0.5 (0.3)	0.5 (0.2)
	Rotation	**7.5 (2.1)**	5.3 (2.0)	8.6 (2.4)	5.1 (2.5)	0.2 (0.3)	0.6 (0.3)	0.4 (0.3)	0.6 (0.3)

**Table 2 sensors-22-03259-t002:** Values between calibration methods and MoCap during the different motions for knee joint. ∆ROM: absolute difference in range of motion, ∆Peak: absolute difference in max peak, SD: standard deviation. Significant differences *p* < 0.05—values in bold. W: walking; R: running; S.A. stair ascent; S.D: stair descent.

Knee		ΔROM (°)				ΔPeak (°)			
		Method 0	Method 1	Method 2	Method 3	Method 0	Method 1	Method 2	Method 3
		Mean (SD)				Mean (SD)			
W	Flexion	**3.9 (1.7)**	0.6 (1.6)	1.7 (1.5)	0.5 (0.9)	6.7 (4.6)	7.2 (4.1)	1.5 (3.9)	0.9 (3.8)
	Adduction	**7.7 (5.0)**	3.4 (4.4)	3.2 (4.2)	1.9 (2.1)	5.9 (4.6)	2.1 (2.6)	3.0 (2.4)	2.9 (2.4)
	Rotation	3.4 (3.5)	3.0 (2.5)	3.4 (1.9)	2.8 (1.8)	**7.5 (5.4)**	6.5 (3.2)	6.4 (3.3)	4.3 (2.5)
R	Flexion	4.8 (2.1)	4.7 (1.9)	5.1 (1.7)	4.9 (2.1)	6.6 (2.3)	5.1 (1.9)	3.2 (1.2)	2.9 (0.8)
	Adduction	5.4 (3.3)	1.5 (1.0)	3.6 (1.0)	2.9 (1.2)	7.6 (3.6)	2.0 (1.4)	4.2 (1.1)	3.8 (1.4)
	Rotation	6.3 (1.2)	5.7 (1.8)	4.8 (1.8)	3.5 (3.0)	5.1 (2.7)	6.8 (3.2)	6.2 (3.0)	3.5 (2.2)
S.A.	Flexion	7.0 (2.1)	5.5 (2.6)	5.3 (2.5)	5.1 (1.8)	6.6 (2.3)	5.1 (1.9)	3.2 (1.2)	2.9 (0.8)
	Adduction	6.4 (3.3)	5.5 (1.0)	6.6 (1.0)	5.9 (1.2)	**7.6 (3.6)**	2.0 (1.4)	4.2 (1.1)	3.8 (1.4)
	Rotation	5.3 (1.2)	5.0 (1.8)	2.8 (1.8)	2.5 (1.4)	9.1 (2.7)	5.8 (3.2)	9.4 (3.0)	4.5 (2.2)
S.D.	Flexion	6.9 (2.1)	3.5 (2.4)	3.1 (2.5)	3.2 (1.8)	7.2 (2.3)	4.7 (1.92	2.3 (1.2)	2.4 (0.8)
	Adduction	4.2 (3.3)	3.6 (1.3)	2.1 (1.7)	1.9 (2.7)	6.5 (2.6)	3.8 (1.4)	4.0 (1.1)	3.6 (1.4)
	Rotation	4.6 (1.5)	3.3 (1.8)	3.6 (1.4)	2.3 (2.4)	8.8 (2.7)	5.2 (2.2)	9.0 (3.0)	6.1 (2.1)

## Data Availability

The data that support the findings of this study are available from the corresponding author upon request.

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
