# Peer review of "Inertial Sensor-to-Segment Calibration for Accurate 3D Joint Angle Calculation for Use in OpenSim"

_sensors, 2022, doi:10.3390/s22093259_

Round 1

Reviewer 1 Report

The manuscript describes a method for sensor-to-segment calibration of inertial capture systems to be used in MSK 3DOF models hip and knee joints. The method described have a better performance that methods found in the literature. 

The topic is interesting as this method can help in the advance of InCap use for motion analysis. The structure of the manuscript is right, whoever, materials and methods, results, and conclusions need to be improved. furthermore, the general readability must bee improved, some sentences are hard to follow  

Comments: 

Methods need to be improved. Provide the position of the MoCap markers, and the InCap sensors, where were they placed  and why? how where they placed to prevent soft tissue motion artifacts and drifts?  Are these effects taken into account in the analysis. The need to be considered. 
Where the time for each step recorder, the frequency of the movements need to be controlled to improve the analysis on both system, do they have to do it in specific times, if yes, how did you secure the right timing of each movement? 

P4L13 iii) is the calibration (i) same as the (iv), for (v), how did you secure that there was no movement in knee or ankle joints, how were these joints constrained. 
(vi) were these task part of the calibration? 
P4L187. why the angular velocity on knee and ankle joint motion can be assumed to be approximately zero. maybe rephrase these sentence. 
P5L235 to L 140. Comparing M3 to M0, RMSE reduced on average ...  here is not clear which one had more reduction, please improve these lines to make clear which method was better. Also, is not clear how InCap compares to MoCap. Authors state that the RMSE decreased , however, this did not decreased, but it was smaller with a different method. 
P5L241, M2 showed a higher RMSE than M1, there was no "decrease". 
Figure 1: adding a legend with the lines colours will improve readability

in the results: when describing each joint it is not clear, it would be good to point out which results are from inCap and compare o MoCap 

The outcome seem fine, but authors need to mention what and why M3 can be better suited for inCap calibration and evaluation, what characteristics and parameters used makes this a better method? 

P10L411; "The method was not described investigated" these line need improvement. 

Reviewer 2 Report

 **** GENERAL COMMENT: The paper is largely experimental, based on a nice and appropriate variety of measurement instruments. The results can be qualified as preliminary, on the way towards an easy-to-use method in less controlled environments than the lab, avoiding the use of MOCAP. The results are preliminary because some important experiments have not yet been done, as mentioned by the authors themselves in the discussion section (e.g. only healthy subjects are considered, the influence of sensor placement and orientation on the measurements have not been studied quantitatively, as could be expected from a journal paper, limitations on the motion due to pathology are discussed but not verified experimentally). The key novelty of the paper is the sensor-to-segment calibration (M3) method, a combination of existing methods (M1 and M2). The paper is addressing specialists in the field of motion analysis and 3D joint kinematics. Also, some of the statements in the discussion section concerning usability for OA patients are not supported by experiments on such patients. In summary: the paper treats a very interesting subject, but it would be more appropriate to submit it as a conference paper. The paper is well-written and easily understandable.

Some experimental results should be added to characterize the sensitivity to sensor placement and to specify the measurement protocol. In case access to OA patients is difficult, the conclusions on usability for patients should be formulated more carefully.

Some more punctual remarks:

**** The introduction section 1 puts emphasis on the OA related populations compared to healthy persons, which illustrates the possible clinical utility of the method, but that is not verified in the experiments that are carried out on solely 11 healthy subjects.

- In line 88, a correlation coefficient is introduced, please specify at first mention “correlation between what?”

- In line 118,  a magnetic field is popping up. It would be appropriate to add a few lines on magnetic sensors, which are mentioned nowhere else.

- In section 2.3, line 170: time intervals of 1 % are mentioned. What do you mean by that?

**** M1 and M2 are supported by references to the literature, but when it comes to the newly developed M3, only a “combination of M1 and M2” is mentioned. It would be appropriate to spend a paragraph or figure on the definition of “combination”, since M3 is the main novelty of the paper.

- In section 3, Line 232: “M0 (8 degrees),… M3 (2 degrees)”. What are these values exactly?

- Figure 1: the caption seems incomplete to me  -- > What are the 3 columns representing (which planes)?

****- In section 4, lines 368-371: this is a statement that is not coming out of experiments on an OA population (idem lines 392-395)

- In section 4, lines 380-382: I do not understand this phrase. Please reformulate it.

Reviewer 3 Report

The authors present a new method that appears to be clinically easy to use. This method was used and evaluated in a group of eleven healthy subjects.

I consider the manuscript to be well prepared and suitable for publication, and I have no complaints.

I just give two points to consider:

  • characteristics of probands: it would be appropriate to state why 11 people and a different number of men and women were chosen; there is a typo for body height values - it should probably be 1.77 ± 0.9 m
  • table 1 is very large, too wide and thus very small data, so I consider whether to divide the table into 2 parts 

Round 2

Reviewer 1 Report

The authors addressed previous comments,

just need to address some errors and a further comment before publishing:

P12L413 - “ Furthermore, the repeatability and the effect of sensor placement of the measurements in this study was not investigated” this line is not enough. authors need to mention what could be the effect of not analysing repeatability and placement, and how this might have affected their results. 

Reviewer 2 Report

The authors have addressed the different comments and solved some remarks by using more nuance in the text, so that some discussion items can remain open for future work. I acknowledge the fact that the study has been executed with care and that it represents a large amount of experimental work. The reply to some of my questions would require substantially more experimental work, and  hence I can accept the responses given.

A few minor remarks: authors should be aware that Sensors is a Journal for a broader audience than only the experts in their application field. Therefore, I propose to add in the manuscript some elements mentioned in the answers to the questions but not taken up in the paper.

Line 120: magnetic distortions [9] -- > magnetic distortions that may affect the IMU sensors orientation

Line 171:   time intervals of 1% ---  > time intervals of 1 % of a gait cycle

Line 241: plane offset compared to the other methods: M0 (8°), M1 (6°), M2 (4°), M3 (2°) -- > average plane offset compared to the other methods: M0 (8°), M1 (6°), M2 (4°), M3 (2°). These offsets with respect to MoCap angular values can be noticed at the start of the curves in Figure S2.
